# Enrichment of circulating trophoblasts from maternal blood using filtration-based Metacell® technology

Jana Weymaere[1], Ann-Sophie Vander Plaetsen[1], Yasmine Van Den Branden[2], Eliska Pospisilova[3], Olivier Tytgat[1], Dieter Deforce[1], Filip Van Nieuwerburgh[1]*

**1** Laboratory of Pharmaceutical Biotechnology, Ghent University, Ghent, Belgium, **2** Obstetrics and gynecology, Ghent University Hospital, Ghent, Belgium, **3** Faculty Hospital Králové Vinohrady, Laboratory Diagnostics, Prague, Czech Republic

* Filip.VanNieuwerburgh@UGent.be

**Data Availability Statement:** All relevant data, including raw data (Cq values) are contained within the paper and its Supporting Information file.

## Abstract

In a cell-based non-invasive prenatal test (cbNIPT), intact circulating trophoblasts (CTs) are isolated from maternal blood for subsequent genetic analysis. Enrichment of these CTs from maternal blood is the most challenging step in the cbNIPT workflow. This study aims to assess the suitability of the filtration-based Metacell® technology to enrich CTs from maternal blood at week 10 to 13 of gestation. The Metacell® technology is a novel size-based enrichment technology that combines blood filtration through 8 μm pores with an *in vitro* culture method. Three protocols were evaluated. First, 8 mL or 16 mL of maternal blood was filtered and subsequently cultured *in vitro* on the separation membrane for 3 days in RPMI 1640. In addition, 16 mL of maternal blood was filtered, and immediately processed without further culturing. Y-chromosome-specific qPCR or STR analysis was performed to evaluate the enrichment of CTs. A total of 44 samples from pregnant women, out of which 26 were carrying a male fetus, were processed. Although five enriched male fetus samples show detectable male DNA quantities, it cannot be excluded that the obtained positive signal is caused by cell-free fetal DNA sticking to the Metacell® separation membrane. In conclusion, the Metacell® technology, tested as described, is not suitable for consistent enrichment of CTs.

## Introduction

The field of non-invasive prenatal screening has evolved tremendously in the past few years. The implementation of cell-free non-invasive prenatal testing (cfNIPT) in clinical practice resulted in a significant reduction of invasive prenatal testing [1]. cfNIPT is based on massively parallel sequencing of cell-free DNA (cfDNA) in maternal plasma, which is a mixture of maternal and fetal cfDNA (cffDNA) [2–5]. In addition to the detection of large copy number variants and chromosomal aneuploidies, which has been widely adopted in the clinic, some monogenic diseases can be detected using a targeted sequencing approach [4–6]. However, a number of relevant genetic disorders caused by small genetic aberrations are often missed [7–9].

**Funding:** This study was funded by two PhD grants from the Special Research Fund (BOF) of Ghent University awarded to Jana Weymaere(J.W.) (BOF17/DOC/265) and Olivier Tytgat (O.T.) (BOF18/DOC/200). URL BOF: https://www.ugent.be/en/research/funding/bof/overview.htm J.W. did the set-up, the practical part, data acquisition, data analysis, data interpretation, and wrote the main manuscript text. O.T. participated in study design and conceptions and gave critical revisions.

**Competing interests:** We confirm that this manuscript has not been published elsewhere and is not under consideration by another journal. All authors have approved the manuscript and agree with submission to PLOS ONE. The authors have declared that no competing interests exist.

An alternative approach is a cell-based NIPT (cbNIPT) in which intact circulating fetal cells are isolated from maternal blood and subsequently analyzed. A sensitive and accurate cbNIPT would allow the detection of smaller genetic aberrations than cfNIPT, as the isolated fetal cells provide a pure source of fetal genomic DNA without a disturbing background of maternal DNA [10]. Circulating trophoblasts (CTs) and nucleated red blood cells are frequently targeted fetal cells in cbNIPT research [11]. Trophoblasts are cells of the placenta. The terms 'fetal DNA' and 'fetal cells' used throughout this paper refer to these trophoblasts, and it should be considered that the genomic make-up of this 'fetal DNA' can differ from that of the fetus due to the possible presence of confined placental mosaicism. Because of to the larger cell size of CTs [12] compared to other blood cells and the availability of relatively specific cell markers [13,14], most cbNIPT research nowadays focuses on CTs. In a general beginning-to-end work-flow for cbNIPT, maternal blood is collected at week 10 to 13 of gestation. CTs present in maternal blood are then enriched, stained, and imaged. Next, CTs are selected based on staining and/or morphology and subsequently isolated using a single cell isolation technique. Eventually, after whole genome amplification, prenatal testing is performed by analyzing fetal DNA. Although this workflow for cbNIPT is technically challenging due to the rarity and fragility of CTs in the maternal peripheral bloodstream [15], some research groups have already made significant progress in the quest for a cbNIPT.

The Danish research group ARCEDI Biotech (Vejle, Denmark) succeeded in enriching CTs using magnetic-activated cell sorting [16–18]. Likewise, other research groups published evidence for the feasibility of a cbNIPT using a marker-based enrichment strategy [19–21]. Although the results are promising, the major drawbacks of these marker-based strategies are the required volume of maternal blood, the need for pre-enrichment steps such as red blood cell lysis and cell fixation, and their complex and time-consuming procedures.

Instead of marker-based CT enrichment, size-based enrichment could be a valuable alternative in a cbNIPT workflow. Due to the similar physical characteristics of CTs and circulating tumor cells (CTCs) [12], size-based enrichment technologies that have been optimized for the enrichment of CTCs in cancer liquid biopsy, such as Screencell [22], Parsortix [23], and Isolation-by-Size of Epithelial Tumor cells (ISET) [12], could potentially be applied for CT enrichment. The research group of Paterlini-Bréchot reported the successful enrichment and isolation of one to two CTs per mL, starting from 10 mL of maternal blood [24], using the ISET technology. In this technology, blood is first diluted with a proprietary buffer that lyses erythrocytes and fixes nucleated cells. The diluted blood is then filtered under negative pressure through a filter containing pores of 8 µm. As a result, larger cells in the blood, such as CTCs and CTs, are enriched on the filter. Cells with a size of 15 µm or larger are then selected and isolated using laser capture microdissection [24]. Although the publication of Mouawia *et al.* [24] in 2012 seemed very promising, no additional papers using ISET for CT enrichment have been published since then. In addition to this size-based filtration enrichment, successful size-based microfluidic enrichment of CTs is also reported in the literature [25,26]. However, these published technologies still need to be further developed before they can be implemented in a cbNIPT workflow.

The present study investigates the Metacell® technology (Metacell® s.r.o., Ostrava, Czech Republic), a novel size-based enrichment technology combined with an *in vitro* culture method, originally developed for the enrichment and culture of viable CTCs. In this Metacell® technology, fresh blood is filtered through a polycarbonate membrane with pores of 8 µm in less than 10 min. The filtration flow is driven by capillary forces induced by an absorbent touching the separation membrane [27], as visualized in Fig 1A. Due to the gentle flow in this separation process and the absence of pre-enrichment steps, enriched cells remain viable and intact. Therefore, cells on the separation membrane are suitable for culturing, characterization, and cytomorphological evaluation [27]. Fig 1B represents the Metacell® procedure.

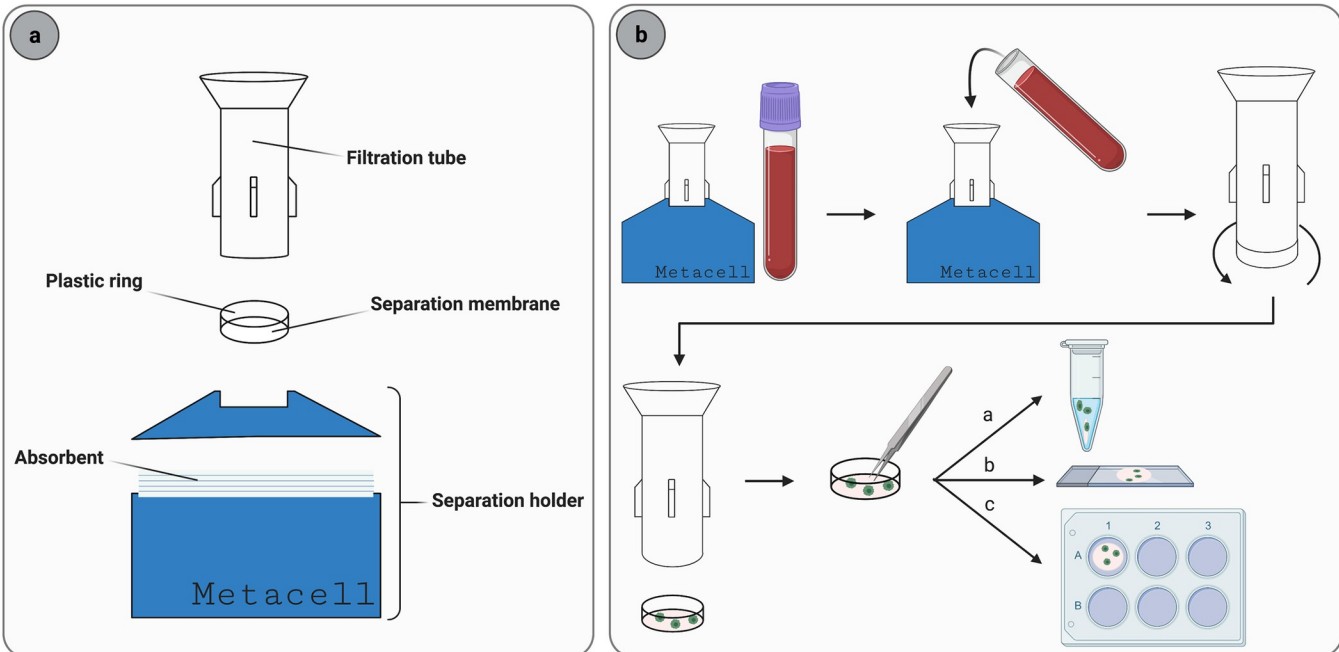

**Fig 1. Overview of the Metacell® technology. A)** Metacell® device in detail. **B)** Metacell® procedure. Blood is collected in the filtration tube with separation membrane. To start the filtration, the filtration tube is pushed down to make contact with the absorbent underneath. After filtration, the plastic ring with the separation membrane is unscrewed from the filtration unit. The separation membrane is then detached and used for characterization (a), cytomorphological evaluation (b), or culturing (c) followed by characterization or cytomorphological evaluation.

Besides successful CTC enrichment [27–29], the Metacell® technology has already proven to be effective for the enrichment of rare circulating endometrium cells (CECs) [30–32]. In this study, the suitability of the Metacell® technology for the enrichment of large fetal cells from maternal blood at week 10 to 13 of gestation is assessed. As trophoblasts are the largest and most abundant type of circulating fetal cells [11], it is most likely that these are enriched from maternal blood using size-based methods. If effective, the Metacell® technology could easily be implemented in a cbNIPT workflow, as visualized in Fig 2.

## Materials and methods

### Experimental design

The suitability of the Metacell® technology was assessed for the enrichment of CTs from maternal blood at 10 to 13 weeks of gestation. To facilitate this assessment, mainly blood samples from women carrying a male fetus were processed, followed by Y-chromosome-specific detection. Processing maternal blood from women carrying a female fetus requires autosomal STR-analysis. However, the interpretation of the result would be much more challenging, as at least 50% of the autosomal STRs overlap between mother and fetus. Moreover, the limited fraction of fetal cells in the presence of maternal cells results in a huge allelic imbalance which hampers efficient amplification and detection of the minor allele fraction. In order to avoid these challenges, mainly blood samples from women carrying a male fetus were processed in this proof-of-concept study.

As visualized in Fig 3, three protocols that vary in maternal blood volume and Metacell® procedure, were evaluated. In the first protocol, which has already proven to be successful for the enrichment of CTCs and CECs [27–32], 8 mL of maternal blood was filtered through a Metacell® separation membrane. Assuming that culturing will result in an additional

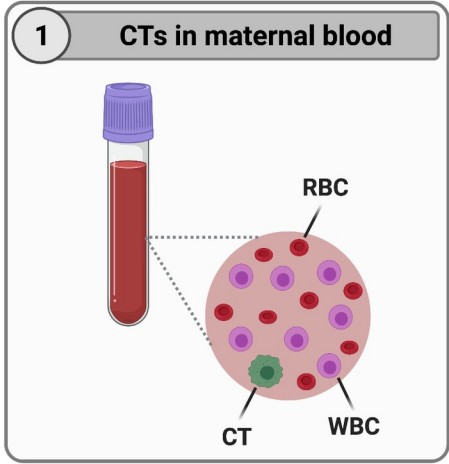
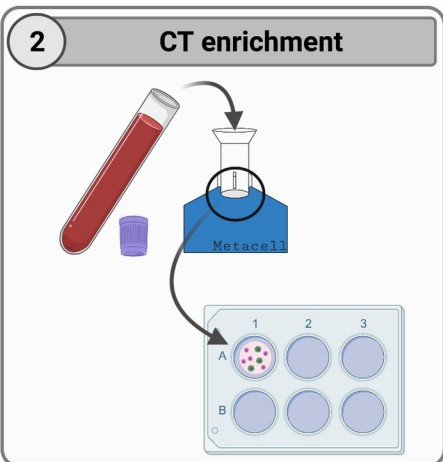
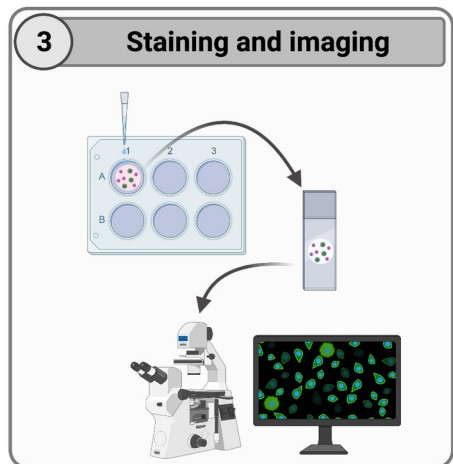
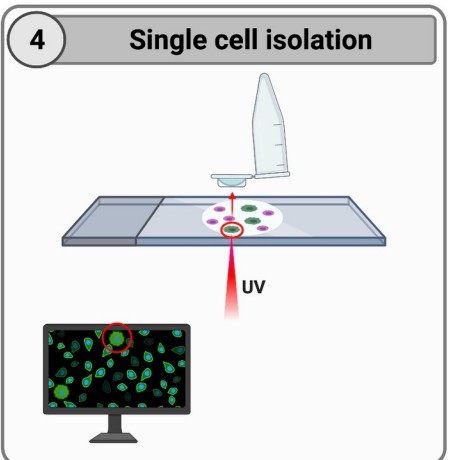
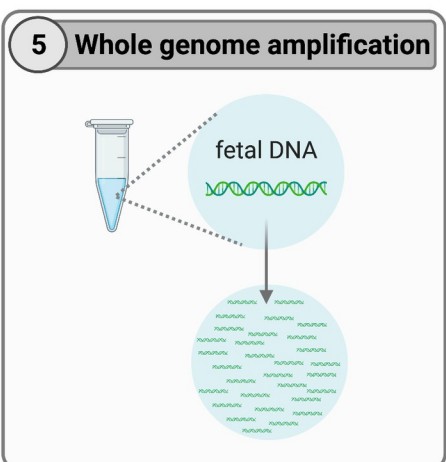
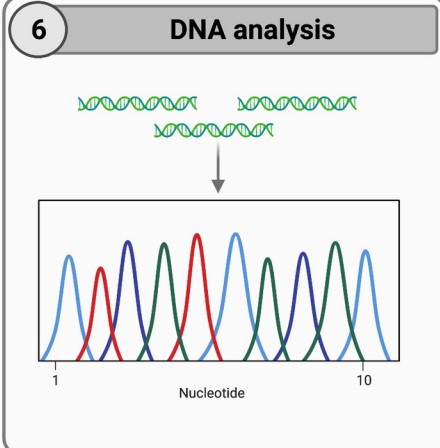

**Fig 2. Metacell Ⓡ CT enrichment and related procedures implemented in a cbNIPT workflow.** (1) Maternal blood containing a low fraction of CTs, is collected at week 10 to 13 of gestation. (2) CTs are enriched using the size-based Metacell Ⓡ technology. Larger cells, such as CTs, remain on the separation membrane. Optionally, the membrane with cells is further cultured to deplete more unwanted cells. (3) A fluorescent dye is added to the culture medium. Cells on the membrane are then visualized with an LCM fluorescence microscope. (4) Based on cytomorphology, cytochemical staining, or immunostaining, a single CT is selected and isolated with UV-LCM. (5) The genome of the single cell is then amplified. (6) Eventually, DNA analysis is performed.

enrichment, the separation membrane was subsequently cultured in Rosewell Park Memorial Institute 1640 medium (RPMI) supplemented with L-glutamine, FBS, and a mix of penicillin and streptomycin for 3 days. In the second protocol, 16 mL of maternal blood was filtered, followed by culturing for 3 days. To assess the effect of culturing, in the third protocol, 16 mL of maternal blood was filtered through a Metacell Ⓡ membrane, and the DNA was extracted without a culturing step.

Following the Metacell Ⓡ procedure, the presence of male cells on the separation membrane was assessed by Y-chromosome-specific detection. As only small amounts of male cells are expected on the Metacell Ⓡ membrane, Y-chromosome short tandem repeat (Y-STR) analysis was chosen for detection because it has a high sensitivity. However, using Y-STR analysis, the obtained signal cannot be quantified. Therefore, in a minority of samples, Y-chromosome quantitative PCR (Y-qPCR) analysis was used in this proof-of-concept study, as Y-qPCR allows quantification. In each assay, a few maternal blood samples from women carrying a female fetus were processed in parallel to serve as a biological negative control.

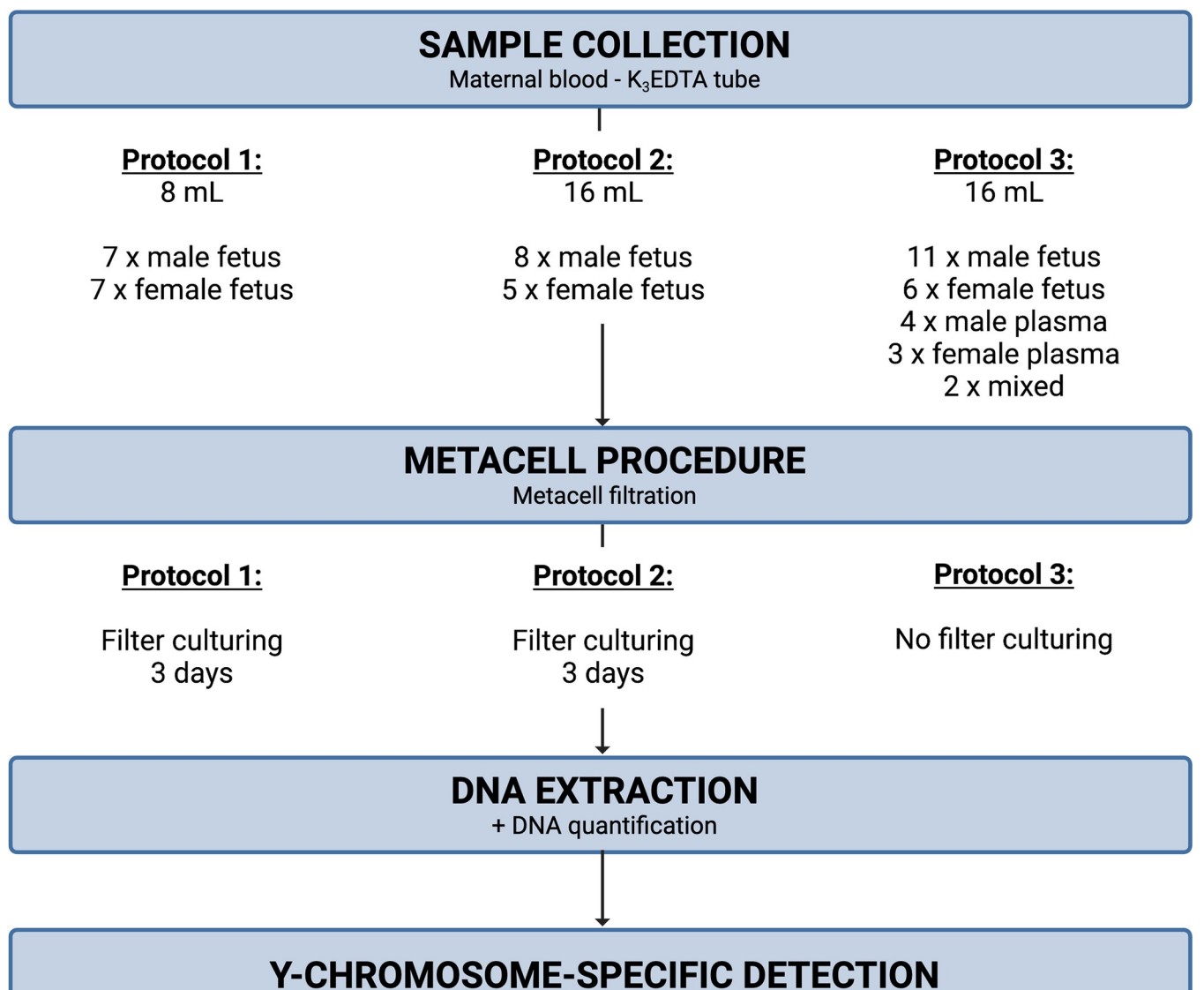

**Fig 3. Experimental design.** Three protocols with varying maternal blood volumes and Metacell® filtration procedures were evaluated in thisCle proof-of-concept study. Each protocol comprises sample collection, Metacell® procedure, DNA extraction, and Y-chromosome-specific detection.

cffDNA present in the sample may possibly stick to the separation membrane during filtration, causing a Y-qPCR or Y-STR signal that is not caused by male fetal cells being retained on the membrane. To assess this assumption, plasma samples from pregnant women, which contain by definition cffDNA without fetal cells, were processed according to protocol 3. Plasma from four male fetus blood samples and three female fetus blood samples was filtered. In addition, to include the possible influence of blood components on the membrane, two samples containing male cffDNA in a blood matrix without male fetal cells were prepared by replacing the plasma from a female fetus blood sample by plasma from a male fetus blood sample. The latter samples are further referred to as 'mixed samples'.

## Sample collection

A total of 55 maternal blood samples were collected from women who attend their gynecologist at Ghent University Hospital for a cfNIPT at 10 to 13 weeks of gestation. All women were adult, healthy, and have signed an informed consent. The study was approved by the ethical review board of Ghent University Hospital (B670201524235), and performed in accordance with the Declaration of Helsinki.

Each maternal blood sample contained 18 mL of peripheral blood, collected in $K_3$EDTA blood collection tubes (ThermoFisher Scientific, Waltham, USA). As the sex of the fetus was not known at the time of collection, a Y-chromosome-specific qPCR was performed on an aliquot of the samples as follows: within 2 hours after collection, 2 mL of blood was centrifuged at 1,200 x g for 12 min. After centrifugation, 400 µL plasma was collected, and cell-free DNA was extracted in 100 µL nuclease-free water using the QIAamp DNA Blood Mini Kit (Qiagen, Hilden, Germany). Next, a qPCR was performed to detect the Y-chromosome-specific multicopy *DYS14* sequence located within the *TSPY* gene, as proposed by Picchiassi *et al.* [33]. Amplification occurs in a total volume of 50 µL, containing 30 µL extracted DNA, 0.9 µM of both forward (5' `GGGCCAATGTTGTATCCTTCTC`) and reverse (5' `GCCCATCGGTCACTTA CACTTC`) primer (Integrated DNA Technologies, Leuven, Belgium), 2.5 U HotStarTaq DNA Polymerase (Qiagen, Hilden, Germany), 1.25 µM EvaGreen dye in aqueous solution (VWR, Radnor, USA), 1 X PCR buffer (Qiagen, Hilden, Germany), and 200 µM of each dNTP (ThermoFisher Scientific, Waltham, USA). The qPCR was performed in triplicate using a LightCycler 480 (Roche, Basel, Switzerland) with cycling conditions of 15 min at 95˚C, followed by 50 cycles of 95˚C for 30 s, 57˚C for 45 s, and 72˚C for 1 min. Each run included two negative controls, consisting of 30 µL sterile nuclease-free water and 30 µL purified plasma from women carrying a female fetus, and a positive control, consisting of 1 ng 2800M male Control DNA (Promega, Madison, USA). Quantification cycle ($C_q$) values were determined with LinRegPCR version 2020.0 [34]. If plasma $C_q$ values were below 35 (S1 Table), the pregnant woman was considered to carry a male fetus. The fetal sex was later confirmed by the Ghent University Hospital, based on cfNIPT results. All women had singleton pregnancies, except for one woman who was expecting a twin (sample 18).

## Metacell® procedure

Maternal blood was transferred to the Metacell® filtration tube. Next, to start the filtration process, the filtration tube was pushed down to make contact with the absorbent underneath. After blood filtration, the filter was washed twice with 5 mL Rosewell Park Memorial Institute 1640 medium (RPMI). The plastic ring with separation membrane was then cultured for 3 days in a humidified atmosphere containing 5% $CO_2$ at 37˚C in 4 mL RPMI medium supplemented with 2 mM L-glutamine, 15% FBS, and a mix of penicillin and streptomycin at a final concentration of 100 units/mL and 100 µg/mL, respectively (all from Gibco, ThermoFisher

Scientific, Waltham, USA). Next, the separation membrane was removed from the plastic ring and collected in 400 μL lysis buffer (Qiagen, Hilden, Germany). In protocol 3, the separation membrane was immediately collected in lysis buffer, without culturing the enriched cells.

## DNA extraction and quantification

The Metacell® procedure was followed by DNA extraction of the enriched cells using the QIAamp DNA Blood Mini kit. First, cells on the separation membrane were lysed by incubating the membrane in 400 μL lysis buffer and 40 μL proteinase K at 56˚C for 10 min. Next, 400 μL EtOH 100% was added. After mixing, the supernatant was purified according to manufacturer's recommendations. DNA was eventually eluted in 35 μL or 40 μL sterile nuclease-free water, for samples analyzed with Y-qPCR or Y-STR, respectively. For each protocol, at least three random samples were quantified with a Qubit dsDNA High Sensitivity Assay Kit (Life Technologies, Carlsbad, USA) to evaluate the cell depletion performance. The approximate number of cells remaining after filtration, was calculated by dividing this DNA quantity by 6 pg per human diploid cell. The average fold depletion is then calculated by dividing the average range of white blood cells in 1 mL blood, which is $4.5–11 \times 10^6$ as reported by Dean [35], by the number of remaining cells.

## Y-chromosome-specific detection

After DNA extraction of the enriched cells, the presence of male DNA was assessed with Y-chromosome-specific detection, either using Y-qPCR or Y-STR analysis. Fig 3 shows the exact number of samples assessed with each technology, categorized by protocol and fetal sex.

## Y-chromosome quantitative PCR analysis

Y-qPCR analysis was performed on the extracted DNA to detect the Y-chromosome-specific multicopy *DYS14* sequence within the *TSPY* gene, as previously described in section 3.2 'Sample collection'. Two positive controls containing 1 ng and 10 pg male 2800M Control DNA, and a no template control (NTC), consisting of 30 μL sterile nuclease-free water, were included in each run.

To validate the Y-qPCR assay, a dilution series of male DNA was analyzed, containing an absolute quantity of 1,000 pg, 100 pg, 50 pg, 20 pg, 10 pg, 5 pg, and 2.5 pg male 2800M Control DNA. To mimic the maternal DNA background present in the enriched Metacell® samples, 70 ng female DNA was added to each calibrator sample. This female DNA was obtained from a blood sample of a woman carrying a female fetus, extracted with the QIAamp DNA Blood Mini kit. An NTC consisting of sterile nuclease-free water, and a negative control containing 70 ng female DNA, were included in each run. The $C_q$ values of all calibrator samples, the negative controls, and the NTC samples are shown in S2 Table. The standard curve with $R^2$ and equation is shown in S1 Fig. The calibrator samples were used to determine the limit of detection (LOD) and limit of quantification (LOQ) of the assay. The LOD was defined as the lowest DNA quantity in the calibrator samples for which the $C_q$ value of all replicates falls outside the $C_q$ range of the negative controls containing 70 ng female DNA. The LOD in this assay was 10 pg male DNA. This LOD corresponds to a $C_q$ of 35.2, which is the geometric mean of the $C_q$ values of the 10 pg male DNA calibrator samples. The LOQ, defined as the lowest DNA quantity in the calibrator samples for which the replicates show a coefficient of variation (CV) $\leq$ 35% [36], was 50 pg male DNA, corresponding with a $C_q$ of 32.2. Assuming that each human diploid cell contains 6 pg of DNA, the LOQ corresponds with approximately 8 male cells.

### Y-chromosome STR analysis

Y-chromosome STRs were amplified with the Investigator Argus Y-12 QS Kit (Qiagen, Hilden, Germany), which is a multiplex PCR kit that targets 11 Y-chromosome STR loci. Each reaction contained 33.8 μL purified DNA, 10 μL Reaction Mix A, 5 μL Primer Mix, and 1.2 μL Multi Taq2 DNA polymerase.

Amplification was initiated with a denaturation step at 94˚C for 4 min, followed by 5 cycles of 94˚C for 30 s, 63˚C for 120 s, and 72˚C for 75 s and 28 cycles of 94˚C for 30 s, 61˚C for 120 s, and 72˚C for 75 s. A final elongation step at 68˚C for 60 min was performed. A positive control consisting of 0.5 ng male 9948 Control DNA (Qiagen, Hilden, Germany), and an NTC consisting of 33.8 μL sterile nuclease-free water, were included for quality control. After capillary electrophoresis of the STR amplicons with the ABI3130xl Genetic Analyzer (Thermo-Fisher Scientific, Waltham, USA), STR profiles were acquired and analyzed with the GeneMapper ID-x 1.2 software (ThermoFisher Scientific, Waltham, USA). Male DNA was detected if at least one allele in the Y-STR profile had a relative fluorescence unit (RFU) higher than the detection limit of 50 RFU.

Analogous to Y-qPCR, the Y-STR assay was validated using a dilution series of male DNA, containing an absolute quantity of 1,000 pg, 500 pg, 100 pg, 50 pg, 20 pg, and 10 pg male DNA in a background of 70 ng female DNA. An NTC consisting of sterile nuclease-free water, and a negative control, containing 70 ng female DNA, were included in each run. The Investigator Argus Y-12 QS Y-STR assay kit was developed to detect male DNA in a mixture of male and female DNA up to a ratio of 1:4,000. According to the protocol recommendations, the optimal input DNA quantity under standard conditions is 0.2–0.5 ng, although reliable results can be obtained with less than 0.1 ng input DNA. As the lowest calibrator sample contains 10 pg male DNA in a background of 70 ng female DNA, an incomplete Y-STR profile is expected. Nevertheless, all calibrator samples show at least 9 of the 11 alleles. The STR profile of a calibrator sample containing 10 pg male DNA in a background of 70 ng female DNA is shown in S2 Fig. No male DNA was detected in the negative control and the NTC sample.

## Results and discussion

### Cell depletion performance of the Metacell® technology

The Metacell® procedure vastly reduces the number of remaining maternal cells. While other successful CT enrichment technologies result in a 100 to 1,000-fold depletion of white blood cells [18,37,38], the Metacell® technology shows a 10,000 fold depletion. Moreover, culturing introduces an additional depletion, reflected in the average fold depletions for protocols 1,2, and 3 shown in Table 1. During culturing, white blood cells may detach from the filter and are washed away, or they may not survive three days culturing. Nevertheless, this excellent depletion performance is no indicator for the success of CT enrichment, which is discussed in the next section.

**Table 1. Overview of the Metacell® performance.** For each protocol, the average DNA quantity after enrichment, the corresponding number of remaining cells, and the average fold depletion is calculated.

| Protocol | Processed blood volume | Average DNA quantity (ng) after enrichment (SD) | | Corresponding number of remaining cells (SD) | Average fold depletion |
|---|---|---|---|---|---|
| 1 | 8 mL | 7.38 (6.66) | (n = 9) | 1230 (1109) | 2.9–7.1 x $10^4$ |
| 2 | 16 mL | 28.64 (12.79) | (n = 3) | 4773 (2131) | 1.5–3.7 x $10^4$ |
| 3 | 16 mL | 42.16 (16.52) | (n = 6) | 7026 (2757) | 1.0–2.5 x $10^4$ |

SD: standard deviation.

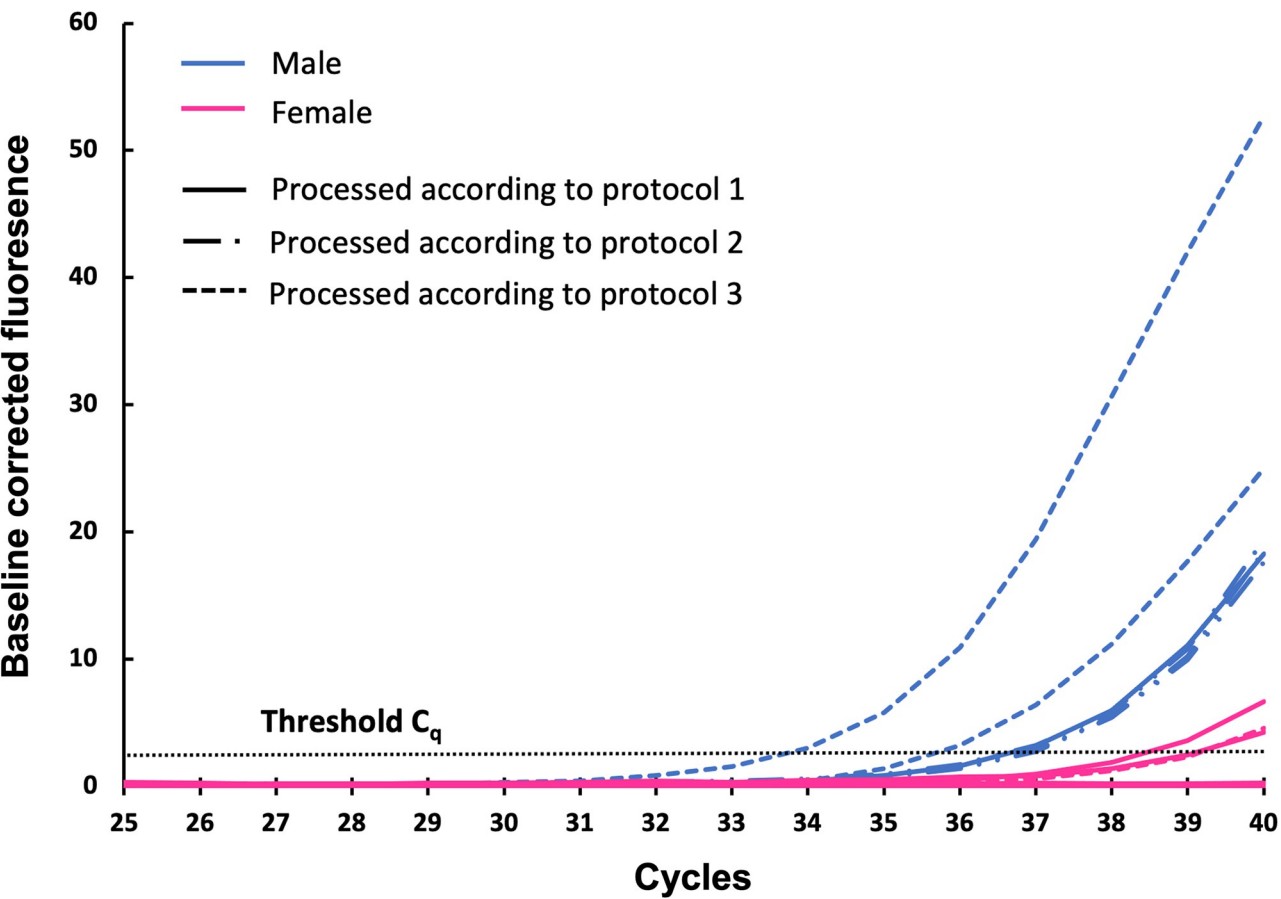

**Fig 4. Baseline corrected fluorescent signal plotted against the cycle number.** Male fetus samples are plotted in blue, while female fetus samples are marked in pink.

### Fetal cell enrichment and detection

A total of 44 blood samples from pregnant women were processed with the Metacell® technology. Of these samples, 26 women were carrying a male fetus, and 18 women were carrying a female fetus. In addition, plasma of seven blood samples was processed and four blood samples were used to create two mixed samples (Section 3.1 'Experimental design').

A total of 27 samples, including 9 male fetus samples, 9 female fetus samples, 4 male plasma samples, 3 female plasma samples, and 2 mixed samples, were assessed using Y-qPCR analysis. In Fig 4, the baseline corrected fluorescent signal of all fetus samples is plotted against the cycle number.

Individual $C_q$ values for all male fetus samples, male plasma samples, and mixed samples, categorized per protocol, are listed in Table 2. $C_q$ values for all female fetus samples and female plasma samples are listed in S3 Table. All samples with a $C_q$ value below the predefined LOD of 10 pg, corresponding to a $C_q$ value of 35.2, are considered positive for male DNA. As a result, all female fetus samples and female plasma samples are negative for male DNA. Male DNA can be detected with Y-qPCR analysis in 1 out of 3 male fetus samples processed according to protocol 3 (sample 18). All male fetus samples processed according to protocol 1 or 2 have $C_q$ values above the LOD. However, although the mixed samples have $C_q$ values above the LOD, 2 out of 4 male plasma samples show detectable amounts of male DNA. This finding

**Table 2. Overview of the Y-qPCR and Y-STR results for male fetus samples, male plasma samples, and mixed samples, categorized by protocol and analysis method.**

| PROTOCOL 1 | | | |
|---|---|---|---|
| Analysis | Sample | Result | Fetal DNA detectable? |
| Y-qPCR | 1 | $C_q > 40$ | No |
| | 2 | $C_q > 40$ | No |
| | 3 | $C_q = 36.87$ | No |
| Y-STR | 4 | Empty | No |
| | 5 | Empty | No |
| | 6 | 1 allele | Yes |
| | 7 | Empty | No |
| PROTOCOL 2 | | | |
| Analysis | Sample | Result | Fetal DNA detectable? |
| Y-qPCR | 8 | $C_q = 37.05$ | No |
| | 9 | $C_q = 37.98$ | No |
| | 10 | $C_q = 37.09$ | No |
| Y-STR | 11 | Empty | No |
| | 12 | 2 alleles | Yes |
| | 13 | Empty | No |
| | 14 | Empty | No |
| | 15 | Empty | No |
| PROTOCOL 3 | | | |
| Analysis | Sample | Result | Fetal DNA detectable? |
| Y-qPCR | 16 | $C_q = 35.54$ | No |
| | 17 | $C_q > 40$ | No |
| | 18 | $C_q = 33.99$ | Yes |
| | Plasma 1 | $C_q = 36.54$ | No |
| | Plasma 2 | $C_q = 34.86$ | Yes |
| | Plasma 3 | $C_q = 36.26$ | No |
| | Plasma 4 | $C_q = 31.19$ | Yes |
| | Mixed 1 | 36.23 | No |
| | Mixed 2 | 37.92 | No |
| Y-STR | 19 | 1 allele | Yes |
| | 20 | Empty | No |
| | 21 | Empty | No |
| | 22 | 2 alleles | Yes |
| | 23 | Empty | No |
| | 24 | Empty | No |
| | 25 | Empty | No |
| | 26 | Empty | No |

$C_q$: quantification cycle, Y-qPCR: Y-chromosome-specific quantitative PCR, Y-STR: Y-chromosome-specific short tandem repeat.

reduces the evidential value of the positive signal in male fetus samples as originating from intact fetal cells retained on the membrane. The positive signal may not only be caused by the actual presence of male CTs, but also by male cffDNA.

Remarkably, positive sample 18 originates from a twin pregnancy (boy and girl). Although the $C_q$ of this sample is above the $C_q$ corresponding to the LOQ, the male DNA quantity in

sample 18 is estimated as 19.69 pg, which corresponds to approximately 3 male cells. This finding can be a coincidence due to random biological variation in the number of CTs or cffDNA levels, or due to physiological differences between singleton and twin pregnancies at gestational week 10 to 13. However, Ravn *et al*. [15], and Van de Looij *et al*. [39] did not observe any apparent changes in CT numbers in twin pregnancies compared to singleton pregnancies, based on a few cases. Besides, there is no clarifying literature available on the cffDNA fraction in twin pregnancies in gestational week 10 to 13.

Y-STR analysis was performed on a total of 26 samples, including 17 male fetus samples and 9 female fetus samples. All female fetus samples were negative for male DNA. As presented in Table 2, male DNA could only be detected in 1 out of 4 male fetus samples processed according to protocol 1 (sample 6), 1 out of 5 male fetus samples processed according to protocol 2 (sample 12), and 2 out of 8 male fetus samples processed according to protocol 3 (samples 19 and 22). Their STR profiles are shown in S3 Fig. However, similar to Y-qPCR, the obtained signal is too low to exclude cffDNA detection.

Regardless of the detection method used, samples processed according to protocol 3 have the highest success rate (3 out of 11 samples, 27%). The culturing step included in protocol 1 and 2 does not only result in depletion of unwanted cells, but may also have a negative influence on target cell recovery.

Overall, our results reveal that the main concern of the tested set-ups, is the lack of enriched fetal cells, and not the number of contaminating maternal cells. An unfeasible large blood volume should be processed to robustly get the required number of fetal cells for cbNIPT using the protocols as tested here. Several other set-ups could have been tested. For example, we've also processed some maternal samples that were fixed prior to filtration. This was unsuccessful, as the filter was immediately clogged. The results obtained after Metacell® enrichment are in contrast to the reported results obtained with the similar, but successful ISET technology [24], which mainly differs from Metacell® in terms of pre-enrichment and method of filtration flow generation. In ISET, cells are fixed/lysed prior to filtration under negative pressure, while Metacell® has the advantage of filtering whole blood driven by capillary forces. However, unfixed CTs may be deformable, whereby they are filtered through the 8 μm pores, even if they are larger than 8 μm. This might explain why the Metacell® technology is not suitable for implementation in a cbNIPT workflow.

## Conclusion

The Metacell® technology, already proven to be successful for the enrichment of CTCs and CECs, is not suitable for consistent enrichment of CTs using the protocols as tested here. Although five male fetus samples show low, but detectable male DNA quantities, it cannot be excluded that the obtained signal is caused by cffDNA sticking to the separation membrane.

## Supporting information

**S1 Fig. Standard curve, $R^2$, and equation for Y-qPCR analysis.** All calibrator samples consist of male DNA in a background of 70 ng female DNA.
(DOCX)

**S2 Fig. Y-STR profile of a calibrator sample containing 10 pg male DNA in a background of 70 ng female DNA.**
(DOCX)

**S3 Fig. Y-STR profile of sample 6, 12, 19, and 22.** An allele is called if the fluorescent signal is higher than 50 RFU. Called alleles are marked with a blue arrow.
(DOCX)

**S1 Table. Plasma $C_q$ values with corresponding male DNA concentration, and fetal sex confirmation by Ghent University Hospital.**
(DOCX)

**S2 Table. Individual $C_q$ values of the calibrator samples used for Y-qPCR validation, with coefficient of variation.**
(DOCX)

**S3 Table. Individual $C_q$ values of the female fetus samples and female plasma samples, categorized per protocol.**
(DOCX)

# Acknowledgments

We would like to thank the midwives of the University Hospital Ghent for the collection of the maternal blood samples.

# Author Contributions

**Conceptualization:** Jana Weymaere, Ann-Sophie Vander Plaetsen, Olivier Tytgat, Filip Van Nieuwerburgh.

**Data curation:** Jana Weymaere.

**Formal analysis:** Jana Weymaere, Ann-Sophie Vander Plaetsen, Olivier Tytgat.

**Funding acquisition:** Jana Weymaere.

**Investigation:** Jana Weymaere, Ann-Sophie Vander Plaetsen, Yasmine Van Den Branden.

**Methodology:** Jana Weymaere, Ann-Sophie Vander Plaetsen, Eliska Pospisilova, Filip Van Nieuwerburgh.

**Project administration:** Jana Weymaere.

**Resources:** Jana Weymaere, Yasmine Van Den Branden.

**Software:** Jana Weymaere.

**Supervision:** Jana Weymaere, Dieter Deforce, Filip Van Nieuwerburgh.

**Validation:** Jana Weymaere.

**Visualization:** Jana Weymaere.

**Writing – original draft:** Jana Weymaere.

**Writing – review & editing:** Ann-Sophie Vander Plaetsen, Yasmine Van Den Branden, Eliska Pospisilova, Olivier Tytgat, Dieter Deforce, Filip Van Nieuwerburgh.

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
