## [Decision Letter · Decision Letter 0]

8 Mar 2022

PONE-D-21-37386Enrichment of circulating trophoblasts from maternal blood using filtration-based Metacell® technology.PLOS ONE

Dear Dr. Van Nieuwerburgh,

Thank you for submitting your manuscript to PLOS ONE. After careful consideration, we feel that it has merit but does not fully meet PLOS ONE’s publication criteria as it currently stands. Therefore, we invite you to submit a revised version of the manuscript that addresses the points raised during the review process.

We look forward to receiving your revised manuscript.

Kind regards,

Kelvin Yuen-Kwong CHAN, Ph.D.

Academic Editor

PLOS ONE

Journal Requirements:

"No"

Reviewers' comments:

Reviewer's Responses to Questions

**Comments to the Author**

1. Is the manuscript technically sound, and do the data support the conclusions?

Reviewer #1: Partly

Reviewer #2: Partly

2. Has the statistical analysis been performed appropriately and rigorously? 

Reviewer #1: No

Reviewer #2: N/A

3. Have the authors made all data underlying the findings in their manuscript fully available?

Reviewer #1: Yes

Reviewer #2: Yes

4. Is the manuscript presented in an intelligible fashion and written in standard English?

Reviewer #1: Yes

Reviewer #2: Yes

5. Review Comments to the Author

Reviewer #1: Summary of the study:

In search for a biomarker-free workflow for the enrichment of circulating fetal trophoblasts (CT) from maternal blood, the authors present results from a study where they use a filtration-based technology called MetaCell to isolate fetal cells from maternal blood cells.

In the study they use different amounts of blood collected from pregnant women carrying male fetuses, run the sample through the device, and confirm their presence of CTs fetal using either Y specific qPCR, or Y-STR analysis. In few cases they show the presence of male Y-chromosome. However, they don’t rule out that the presence of Y-chromosome could be due to the presence of cell-free-fetal DNA, and not necessarily CTs. The Authors conclude that MetaCell technology is not apt for isolating CTs.

General Comments:

Technologies isolating fetal cells from maternal blood and using them for prenatal diagnosis are attractive with a huge potential to be disruptive. A lot of research is being done not only to find new fetal cell markers, but also to exploit their different size and morphology to isolate them from maternal blood. Though the current study presents a negative result, there are some major questions and concerns that this study either misses or avoids addressing. Here are some concerns:

1. Even though the study claims that the presence of Y specific qPCR is evidence of the presence of circulating fetal trophoblasts, no direct evidence is provided to show that the technology isolates fetal trophoblasts, and not any other fetal cell type.

2. The final technology workflow for using fetal cells for prenatal diagnosis is not clear. Even if the presence of fetal cells is confirmed by qPCR, what is the roadmap for isolating individual fetal cells and using them for genetic analyses?

3. Similarly, if not for Y specific qPCR, or STR for male pregnancies, how would the fetal cells originating from female pregnancies be confirmed?

Reviewer #2: The authors have tested the Metacell technology to enrich cells (trophopblasts) from the conceptus, circulating in maternal blood. I 3 different set-ups, the yield was none or poor. Testing this system is relevant, and the observations are as such relevant to others working in this field. However, the discussion should include considerations on whether the test set-up has tested the system sufficiently. I.e. could larger volumes of blood have been used, could the procedure show efficient if the blood was drawn at a later time. Could the filtering be followed by another procedure to further minimize contamination by maternal cells? Perhaps the authors could think of other possibilities

Minor issues:

Line 34. I recognize that the term “fetal DNA” is frequently used for the DNA originating in the conceptus, identified in the maternal blood. However, please consider using another term, as likely most of this DNA does not originate in the fetus – or at least please comment on this. Similarly, the expression cffDNA is misleading

Line 43. Similarly to the comment to line 34, it would be nice avoid calling trophoblasts “fetal cells” (or at least it would be nice to read a comment on the fact that the circulating cells identified as trophoblasts, more likely originate in the placenta than the fetus).

Line 49: Same comment on “fetal DNA”.

Line 67: “blood cells” should be replaced by a term that indicates that (some of) these cells are not normally present in the blood, e.g. “cells in the blood”

Lines 169 and 170: Some words are missing, likely ”the cells”

Line 176: “supernatans” likely is misspelled

The figures and tables are fuzzy

6. PLOS authors have the option to publish the peer review history of their article (what does this mean?). If published, this will include your full peer review and any attached files.

Reviewer #1: No

Reviewer #2: **Yes: **Lone Sunde

---

## [Author Response · Author response to Decision Letter 0]

11 Apr 2022

Dear reviewers, 

Thank you very much for revising our paper. We appreciate your comments and have considered them for improving our manuscript and resubmitting the paper. All your comments are addressed in a point-by-point response which you can find in the file “Response to reviewers.docx,” which we uploaded together with our revised manuscript. In addition, for your convenience, a “Revised manuscript with Track Changes.docx” has been uploaded alongside a clean version of the revised manuscript.

Kind regards,

Filip Van Nieuwerburgh

---

## [Decision Letter · Decision Letter 1]

27 Jun 2022

Enrichment of circulating trophoblasts from maternal blood using filtration-based Metacell® technology.

PONE-D-21-37386R1

Dear Dr. Van Nieuwerburgh,

We’re pleased to inform you that your manuscript has been judged scientifically suitable for publication and will be formally accepted for publication once it meets all outstanding technical requirements.

Kind regards,

Kelvin Yuen-Kwong CHAN, Ph.D.

Academic Editor

PLOS ONE

Additional Editor Comments (optional):

Reviewers' comments:

Reviewer's Responses to Questions

**Comments to the Author**

1. If the authors have adequately addressed your comments raised in a previous round of review and you feel that this manuscript is now acceptable for publication, you may indicate that here to bypass the “Comments to the Author” section, enter your conflict of interest statement in the “Confidential to Editor” section, and submit your "Accept" recommendation.

Reviewer #1: All comments have been addressed

2. Is the manuscript technically sound, and do the data support the conclusions?

Reviewer #1: Partly

3. Has the statistical analysis been performed appropriately and rigorously? 

Reviewer #1: (No Response)

4. Have the authors made all data underlying the findings in their manuscript fully available?

Reviewer #1: Yes

5. Is the manuscript presented in an intelligible fashion and written in standard English?

Reviewer #1: Yes

6. Review Comments to the Author

Reviewer #1: (No Response)

7. PLOS authors have the option to publish the peer review history of their article (what does this mean?). If published, this will include your full peer review and any attached files.

Reviewer #1: No

---

## [Editor Report · Acceptance letter]

7 Jul 2022

PONE-D-21-37386R1 

Enrichment of circulating trophoblasts from maternal blood using filtration-based Metacell® technology. 

Dear Dr. Van Nieuwerburgh:

I'm pleased to inform you that your manuscript has been deemed suitable for publication in PLOS ONE. Congratulations! Your manuscript is now with our production department. 

Kind regards, 

on behalf of

Dr. Kelvin Yuen-Kwong CHAN 

Academic Editor

PLOS ONE